# Retrospective Detection and Genetic Characterization of *Porcine circovirus* 3 (PCV3) Strains Identified between 2006 and 2007 in Brazil

**DOI:** 10.3390/v11030201

**Published:** 2019-02-27

**Authors:** Giuliana Loreto Saraiva, Pedro Marcus Pereira Vidigal, Viviane Sisdelli Assao, Murilo Leone Miranda Fajardo, Alerrandra Nunes Saraiva Loreto, Juliana Lopes Rangel Fietto, Gustavo Costa Bressan, Zélia Inês Portela Lobato, Márcia Rogéria de Almeida, Abelardo Silva-Júnior

**Affiliations:** 1Laboratório de Infectologia Molecular Animal, Instituto de Biotecnologia Aplicada à Agropecuária, Universidade Federal de Viçosa (UFV), Viçosa, Minas Gerais 36570-900, Brazil; giulianasaraiva@hotmail.com (G.L.S.); jufietto@ufv.br (J.L.R.F.); gustavo.bressan@ufv.br (G.C.B.); marcia@ufv.br (M.R.d.A.); 2Núcleo de Análise de Biomoléculas (NuBioMol), Centro de Ciências Biológicas (CCB), Universidade Federal de Viçosa (UFV), Viçosa, Minas Gerais 36570-900, Brazil; 3Laboratório de Imunobiológicos e Virologia Animal, Departamento de Veterinária, Universidade Federal de Viçosa (UFV), Viçosa, Minas Gerais 36570-900, Brazil; vivi.assao@gmail.com (V.S.A.); muriloleonefajardo@gmail.com (M.L.M.F.); alerrandrasaraiva@hotmail.com (A.N.S.L.); 4Escola de Veterinária, Departamento de Medicina Veterinária Preventiva, Universidade Federal de Minas Gerais (UFMG), Belo Horizonte, Minas Gerais 36570-900, Brazil; ziplobato@gmail.com

**Keywords:** PCV3, phylogeny, epidemiology, evolution, Brazil, retrospective detection

## Abstract

*Porcine circovirus* 3 (PCV3) is an emerging virus that was first identified in the United States in 2016. Since its first detection, PCV3 has already been found in America, Asia, and Europe. Although PCV3 has already been described in Brazil, knowledge of its detection and sequence variation before 2016 is limited, as well as its distribution in the main swine producing regions of Brazil. In this study, 67 porcine clinical samples collected from nine states in Brazil between 2006 and 2007 were analyzed for PCV3 infection by PCR. Results showed that 47.8% of the samples were PCV3 positive, across all nine states. Of the PCV3-positive samples, 37.5% were also positive for PCV2. Interestingly, no clinical signs were associated with samples that were detected singularly with PCV3 infection. Moreover, the positive PCV3 rate in healthy pigs was higher (29.8%) than that found in unhealthy pigs (17.9%), suggesting that most pigs could live with PCV3 infection without any clinical sign in the analyzed samples. Nucleotide sequence analysis showed that PCV3 strains obtained in this study shared 94.44% to 99.83% sequence identity at the open reading frame 2 (ORF2) gene level with available strains from different countries. PCV3 Brazilian sequences collected in 2006 and 2007 shared 97.94% to 99.62% identity with the strains obtained in 2016. The results of neutrality and selective pressure tests indicated that the PCV3 Cap protein seems unable to tolerate high levels of variation on its sequence. Phylogenetic analysis grouped the Brazilian strains in PCV3a and PCV3b genotypes clusters, both including strains collected in America, Asia, and Europe. Taking the results together, multiple events of introduction of PCV3 may have occurred in Brazil, and Brazilian PCV3 strains may show genetic stability over the past 10 years.

## 1. Introduction

Porcine circoviruses (PCVs) are non-enveloped and single-stranded circular DNA viruses. Until recently, PCVs could be separated into two main species: *Porcine circovirus* 1 (PCV1) and *Porcine circovirus* 2 (PCV2) [1]. PCV1 was first discovered in 1974 and was not associated with clinical disease [2,3]. PCV2 was first discovered in the 1990s and is associated with a range of syndromes in pigs called Porcine Circovirus Associated Diseases (PCVAD), which can cause significant economic losses [4,5,6,7]. In 2016, a novel circovirus named *Porcine circovirus* 3 (PCV3) was identified in the United States in sows that died acutely with clinical signs of porcine dermatitis and nephropathy syndrome (PDNS) and were PCV2 negative [8].

Since then, PCV3 was also reported in the Americas, Asia, and Europe and its detection was associated with different clinical presentations in pigs. Reproductive problems were correlated with PCV3 [8,9,10,11,12,13], as well as cardiac and multisystemic inflammation [12], respiratory diseases [14,15,16], congenital tremors in neonatal pigs [17], and diarrhea in weaned piglets [16]. However, the high prevalence of PCV3 in pigs and in wild boars without specific clinical signs has been also reported [18,19,20,21].

Retrospective studies of prevalence were performed to understand the epidemiology and molecular evolution of PCV3. In China, clinical samples of pigs collected between 1990 and 1999 were analyzed and 6.5% were positive for PCV3, with the first cases occurring in 1996 [22]. Results from a retrospective study in the United Kingdom showed that PCV3 was detected in 5% of tissue samples collected between 2001 and 2004 [23]. In Sweden, 20.4% of the samples collected between 1993 and 2007 were positive, and one of these samples was collected in 1993 [19]. Results of an epidemiological study in Spain confirmed that PCV3 has been circulating in the Spanish pig population since 1996 [24]. These results are in agreement with the estimates that PCV3 have emerged over the past 50 years in farmed pigs [9,25].

Two Brazilian genomes of PCV3 were deposited in GenBank, confirming that PCV3 is also circulating in Brazil. The genomes were obtained from pooled serum samples collected in 2016 from sows who had just delivered litters with variable numbers of stillbirths. No PCV3 sequence was detected in samples from sows which had no stillbirths on the same farms [13].

Although PCV3 has already been described in Brazil, knowledge of its prevalence and sequence variation before 2016 is limited, as well as its distribution in the country. The aim of this study was to investigate the epidemiology and molecular evolution of PCV3 from Brazilian pig samples collected before its first detection to answer the following questions: (i) Had PCV3 already circulating in Brazil before 2016? (ii) Can PCV3 be found in the main pork producing regions of Brazil? (iii) How much conserved are the sequences of Brazilian strains? (iv) How significant are the events of introduction of PCV3 in Brazil?

## 2. Materials and Methods

### 2.1. PCV3 Detection

A total of 67 tissue samples as lung, pool of lymph nodes, and spleen were collected between 2006 and 2007 from pigs with clinical signs of PCVAD and in good health conditions, from nine states which are located in the main swine producing regions of Brazil (Midwest, Southeast, and South) (Figure 1). The DNA of these samples was extracted using the Wizard® SV Genomic DNA Purification System following the manufacturer’s recommendations, and the DNA extracted was stored at −80 °C.

The stored DNA samples were submitted to PCR assay for PCV3 detection using primers described previously, which amplify the region that encodes the ORF2 of PCV3 [10]. And another PCR assay for PCV2 detection using the primer forward 5’- CATGCCCTGAATTTCCATATG -3’ and the primer reverse 5’- ATGTTGCTGCTGAGGTGCTGC -3’ that target ORF2 which has been proven to be the major target of the host immune response against PCV2 [26,27].

The 25 μL PCR reaction mixtures contained 12.5 μL of DreamTaq PCR Green PCR Master Mix (2×) (Thermo Fisher Scientific, Massachusetts, EUA), 2 μL of each 10 mM primer, 4 μL of DNA template, and 4.5 μL of H_2_O nuclease free. PCR assays were performed with the following thermal profile: 94 °C for 5 min, 35 cycles of denaturation at 94 °C for 30 s, annealing at 57 °C for 45 s, an extension at 72 °C for 1 min, and a final step of 72 °C for 10 min.

The reactions generated 649 bp PCR products that were subjected to electrophoresis on 1% agarose gel. Positive samples were purified by a GFX^TM^ PCR DNA and Gel Band Purification Kit (GE Healthcare, Little Chalfont, UK) and were submitted to sequencing using the Sanger method by Myleus Biotechnology (Belo Horizonte, Brazil).

### 2.2. Sequence Assembly and Analysis

Chromatograms were analyzed using CLC Genomics Workbench version 8.5.4 (Qiagen). During analysis, sequences were trimmed (quality score limit: 0.01; ambiguous nucleotide residues: 0) and assembled into contigs ranging from 491 to 581 nucleotides using the Assemble Sequences tool.

A dataset of 101 ORF2 sequences of PCV3 isolates originated from different countries and collected between 1996 to 2017 was downloaded from GenBank database (http://www.ncbi.nlm.nih.gov/Genbank). Thus, the final dataset selected contained 108 ORF2 sequences of PCV3, including the seven sequences of Brazilian isolates obtained in this study. The ORF2 nucleotide sequences were aligned using the MUSCLE codon alignment tool of MEGA7 version 7.0.21 [28], with a final alignment of 642 nucleotides.

Phylogenetic tree based on the ORF2 nucleotide sequences was reconstructed using the Bayesian Markov Chain Monte Carlo (MCMC) method in four runs with 10,000,000 generations and a sampling frequency of 1000 implemented in MrBayes version 3.1.2 [29]. The model of nucleotide substitution HKY+G was chosen using jModelTest2 version 2.1.10 [30]. The parameter convergence was analyzed in Tracer version 1.6 (http://tree.bio.ed.ac.uk/software/tracer) and 1% of the generated trees was burnt to produce the consensus tree.

The selection pressures over the ORF2 sequences were estimated using Tajima’s D, Fu and Li’s D, and Fu and Li’s F statistical tests of neutrality which are available in DnaSP version 6.11.01 [31]. These methods evaluate if the number of segregating sites in a sequence alignment departs significantly from the neutral expectation [32,33].

Next, selective pressures acting on each codon of the ORF2 nucleotide sequences were estimated by calculating the difference between non-synonymous (dN) and synonymous (dS) substitution rates per codon using the single-likelihood ancestor counting (SLAC), fixed-effects likelihood (FEL), and internal branches fixed-effects likelihood (IFEL) methods [34], which are available in the Data-Monkey web server (http://www.datamonkey.org/) [35]. Values of dN − dS < 0, =0, and >0 indicate negative selection, neutral evolution, and positive selection, respectively.

## 3. Results

### 3.1. PCV3 Epidemiology Back in 2006 and 2007

Sixty-seven clinical samples collected between 2006 and 2007 from nine states in Brazil were screened for PCV3 by PCR, and 47.8% of the samples (32/67) were found positive. All nine states from the main swine producing regions of Brazil were found positive for PCV3 (Midwest region: Mato Grosso, Mato Grosso do Sul and Goiás states; Southeast region: Minas Gerais, São Paulo, and Espírito Santo states; South region: Santa Catarina, Rio Grande do Sul, and Paraná states). The total positive sample rate of PCV2 was 40.3% (27/67). Of the 32 PCV3-positive samples, 37.5% (12/32) were also positive for PCV2. The singular infection rate of PCV3 in the samples was 29.8% (20/67) (Appendix A).

Interestingly, clinical signs were only identified in PCV2-positive samples or in co-infected samples. Analysis binary was performed, and no correlation was found among PCV3, clinical signs and interaction with PCV2 infection. Moreover, the positive PCV3 rate in healthy pigs was 29.8% (20/67) and, in unhealthy pigs, it was 17.9% (12/67) (Appendix A).

### 3.2. Polymorphism Analysis and Estimation of Selection Pressures

Analysis of ORF2 nucleotide sequences showed that the Brazilian PCV3 strains from 2006 and 2007 (MK060073 to MK060079) share from 97.94% to 99.62% of identity with the Brazilian strains recently collected in 2016 (MF079253 and MF079254), and they share from 94.44% to 99.83% of identity with other PCV3 strains from different countries which were collected between 1996 and 2017. PCV3 strains from China, Denmark, Italy, Korea, and the United States are among those which showed the highest identity with the Brazilian strains, with identities ranging from 98.35% to 99.83%.

Alignment of the ORF2 amino acid sequences revealed that Brazilian PCV3 strains from 2006 and 2007 (MK060073 to MK060079) had up to seven amino acid (aa) substitutions on their sequences when compared with the Brazilian from 2016 (MF079253 and MF079254). Polymorphisms were found in the following aa residues positions of Cap protein: 24 (V24A), 27(K27R), 75 (A75S), 77 (S77G), 100 (T100S), and 104 (F104Y). At position 150, the sequences of strains 916 (MK060074) and 295 (MK060075) had the amino acid Leucine (L), similar to the strain MF079253, while the sequences of remaining strains had the amino acid Isoleucine (I), similar to the strain MF079254 (Table 1).

At position 75 of Cap protein, only the sequence of Brazilian strain 916 (MK060074) and another sequence of a Spanish strain (MG807088) shared the substitution of a Serine (S) in place of Alanine (A), when compared with the set of sequences of PCV3 downloaded from GenBank. At positions 77 and 100, only the sequences of Brazilian strains 1343 (MK060073) and 916 (MK060074) showed a Glycine (G) in substitution to a Serine (S) or Threonine (T) and a Serine (S) in substitution to a Threonine (T), respectively. At position 104, the substitution of a Phenylalanine (F) by a Tyrosine (Y) was found in only the sequences of Brazilian strains 1343 (MK060073) and 872 (MK060076) and other four strains from China and Denmark. Interestingly, this same substitution (F104Y) was commonly found in sequences of strains which were collected in Italy from wild boar.

The tests of neutrality suggest that the evolution of the PCV3 ORF2 gene possibly occurred under the influence of selection and that a purifying selection took place (Table 2), considering the significance and the high negative values of calculated coefficients. The dN/dS ratio is also a versatile way to predict the action of natural selection on gene sequences, once it provides an estimate of selection pressure at the protein level (Kimura, 1983). A global dN/dS ratio of 0.155 was calculated for the ORF2 sequences and up to 17% of its codons were predicted to be evolving under a purifying selection, which indicated a strong purifying selection is acting over the Cap protein (Table 3). Therefore, Cap protein of PCV3 seemed to be subject to strong restrictions and unable to tolerate high levels of variation on its sequence.

### 3.3. Phylogenetic Analysis

Phylogenetic tree of ORF2 sequences revealed at least two clusters which could be defined (named as PCV3a and PCV3b) (Figure 2), both including Brazilian strains and other strains collected in America, Asia, and Europe, and suggesting that PCV3 strains are dispersed worldwide without any geographical component. A third phylogenetic cluster was defined with only four strains from Spain, China, and Germany collected between 2014 and 2017.

Among the Brazilian PCV3 strains sequenced in the present study, strains 1373 (MK060077) and 986 (MK060078) were genetically related to other strains described as being part of the PCV3a cluster, which included the Brazilian strain MG079254. The other Brazilian strains were included in the PCV3b cluster, which included the Brazilian strain MG079253. Brazilian strains 1343 (MK060073) and 872 (MK060076), which showed the amino acid substitution (F104Y), were included in PCV3a cluster and grouped together with the strains obtained from wild boars.

## 4. Discussion

The positive rates of PCV3 were 47.8% in the clinical samples collected between 2006 and 2007 from nine states in Brazil. Analyzing the positive rates in other pig-producing countries, the prevalence of PCV3 in southern China was 21.9%, 27.8%, and 31.1% in 2015, 2016 and 2017, respectively, and the total prevalence of positive samples was 26.7% [9]. Studies in China have reported a positive rate of PCV3 in stillborn, tissue, semen, and serum samples of 34.7% and 59.4% [10,20]. In the United States, a positive PCV3 rate of 12.5% for tissue samples and 55% for serum samples were reported [8]. In Poland, PCV3 was detected in 5.9% to 65% of sera obtained from PCV3-positive farms [18]. In South Korea, the total prevalence of PCV3 in individual oral fluid samples was 44.2% [36]. This wide variation in the prevalence of PCV3 may have been due to the variable number of samples in each study, the epidemic situation of the region, and the intrinsic management of the farms as a level of biosecurity and hygiene.

In the analyzed samples, considering the PCV3 positive samples, the co-infection rate of PCV3 with PCV2 was 37.5% (12/32). And considering all the samples collected, the singular infection rate of PCV3 was 29.8% (20/67). An investigation of the epidemiological characteristics and evolutionary dynamics of PCV3 in southern China found that 22.3% samples were co-infected with PCV2 [9]. Another Chinese study showed a co-infection rate of 45.4% of the PCV3 positive samples [10]. In a screening of 265 clinical samples for co-infection of PCV2 and PCV3, 6.8% of the samples were positive [37]. A co-infection rate similar to that found in our work was noticed in a survey in Shandong Province, China. This study showed a prevalence of 39.3% and 59.4% for PCV3 and PCV2 co-infection and PCV3 infection alone, respectively [20].

In our retrospective study, the positive PCV3 rate in healthy pigs was higher (29.8%) than that found in unhealthy pigs (17.9%), suggesting that most pigs could live with PCV3 infection without any clinical infection sign in the analyzed samples. High positive PCV3 rates in pigs without specific clinical signs was also reported in Poland, China, and Sweden [18,19,20]. In contrast, another survey noticed that the positive PCV3 rate in healthy pigs was lower (19.14%) than that found in unhealthy pigs (37.95%) [38].

Interestingly, no clinical signs of PCVAD were observed in PCV3-positive and PCV2-negative samples analyzed in this study. In an investigation of co-infection of *Torque teno sus virus* 1 (TTSuV1), *Torque teno sus virus* 2 (TTSuV2), and PCV2 with PCV3, no clinical sign in PCV3 positive and PCV2 negative swine were noticed in either multiparous sows or live-born infants [39].

Although the presence of PCV3 DNA was found, our study does not purpose to correlate the detection of PCV3 and its pathogenic potential. For this, requirements would have to have been fulfilled as the explicit description of the clinical disease, combined with a specific histopathological picture and the quantification of the virus in association with the lesions [40]. In our opinion, the presence of PCV3 and the absence of clinical signs may indicate that PCV3-alone is unlikely to cause disease in pigs and other cofactors are required for the clinical manifestation of the disease, as already proven for PCV2 pathogenesis. More applied researches should still be conducted to identify PCV3 pathogenic potential and to better understand the prevalence of PCV3 in Brazil, considering that the sampling used in the present study was not probabilistic.

The Brazilian strains were intermingled with other PCV3 strains from different pig producing countries in the phylogenetic tree (Figure 2), without a specific monophyletic cluster for them. This result suggested that PCV3 has not shown a differentiated independent molecular evolution in Brazil, and multiple events of introduction may have occurred.

PCV3 strains obtained in this study shared 94.44% to 99.83% sequence identity at the ORF2 gene level with available strains from different countries. One Brazilian strain, 1343 (MK060073), from the Southwest region of Brazil showed the highest identity (99.83%) with a strain of Denmark (MF805724). Trade of live pigs from Brazil to Denmark has not been reported, whereas a large number of Danish sows have been exported to other countries, including Brazil (trade statistics extracted from: https://comtrade.un.org/data/). Therefore, a possible event of introduction of PCV3 in Brazil could be explained by the export of live pigs from Denmark to Brazil. Another possible event of introduction of PCV3 in Brazil may have occurred through the trade of live pigs with the United States since the Brazilian strain 986 (MK060078) showed high identity (99.81%) with a United States strain collected in 2016 (KX966193). Although the export of live pigs from Brazil to the United States is rare, this route of dispersion cannot be ruled out (trade statistics extracted from: https://comtrade.un.org/data/).

Our PCV3 sequences collected in 2006 and 2007 shared high identity (97.94% to 99.62%) with the Brazilian strains obtained in 2016 [13], indicating the genetic stability of PCV3 Brazilian strains over the past 10 years. This finding can be corroborated by the neutrality and selective pressure test results, in which the Cap protein of PCV3 seems to be unable to tolerate high levels of variation on its sequence. Specific nucleotide and amino acid marker positions in ORF2 may serve for intraspecies classification and genotyping of PCV3 strains into two main phylogenetic clusters, which might be considered as genotypes PCV3a and PCV3b [9,22,40]. The amino acid marker positions in ORF2 (aa 24, 27, 77 and 150) resulted in a specific aa pattern for genotype PCV3a (V K S I) and for genotype PCV3b (A R S/T I/L) [40].

In the phylogenetic tree, the strains 1373 (MK060077) and 986 (MK060078) were included in the PCV3a genotype cluster, and the strains 1343 (MK060073), 916 (MK060074), 295 (MK060075), 872 (MK060076), and 622 (MK060079) were included in the PCV3b genotype cluster. In addition, these Brazilian strains of booth PCV3a and PCV3b genotypes also showed the substitutions which were previously reported at the codons 24, 27, 77, and 150 in the alignment of the ORF2 sequences. These results confirm that both PCV3a and PCV3b genotypes could be found in Brazil since its first detection in 2016.

Interestingly, two Brazilian sequences from the states of Minas Gerais (MK060073) and Goiás (MK060076) were clustered on the phylogenetic tree with the strains obtained from wild boars in Italy [21] and the other four sequences from China and Denmark. All sequences from this clade showed an aa substitution in the position 104 (F104Y). Only one additional aa substitution (codon 156) distinguished the strain “462_Italy_wildboar_2014” (MG978131) from the other PCV3 strains, considering an alignment of 156 aa of the total 214 aa of Cap protein sequence. The study, which investigated the susceptibility of PCV3 in wild boars, suggested a relevant variability and the absence of closely related strains originating from domestic pigs [21]. In contrast, we suggest that wild boar strains are closely related with the domestic pig strains, considering the analysis of amino acid sequence alignment of Cap protein. Therefore, we suggest that the wild boars may play an important epidemiological role as a reservoir of PCV3 for commercial swine. However, this hypothesis will require further investigations to be fully addressed.

## 5. Conclusions

In this study, we analyzed pigs tissue samples collected between 2006 and 2007 in the main producing regions of Brazil. PCV3 was identified in all states from the Midwest, Southeast, and South regions of Brazil. Interestingly, no clinical signs were associated with samples in which only PCV3 was detected. Moreover, the positive PCV3 rate in healthy pigs was higher than that found in unhealthy pigs, suggesting that most pigs could live with PCV3 infection without any clinical infection sign.

The estimation of selective pressures indicated that PCV3 Cap protein seems to be evolving under a strong purifying selection, being unable to tolerate high levels of variation on its amino acid sequence. Furthermore, PCV3 sequences collected in 2006 and 2007 shared high identity with the Brazilian strains obtained in 2016. These results indicated that PCV3 Brazilian strains probably showed genetic stability over the past 10 years. Multiple events of PCV3 introduction in Brazil may have occurred. In the phylogenetic tree, Brazilian PCV3 strains clustered in both PCV3a and PCV3b genotypes clusters, and two of the Brazilian strains were grouped with the strains obtained from wild boars, suggesting that it may have played an important epidemiological role as a reservoir of PCV3 for commercial swine. The results of this study contributed to retrospectively detect new strains and to understand the genetic diversity and molecular evolution of PCV3 in Brazil.

## Figures and Tables

**Figure 1 viruses-11-00201-f001:**
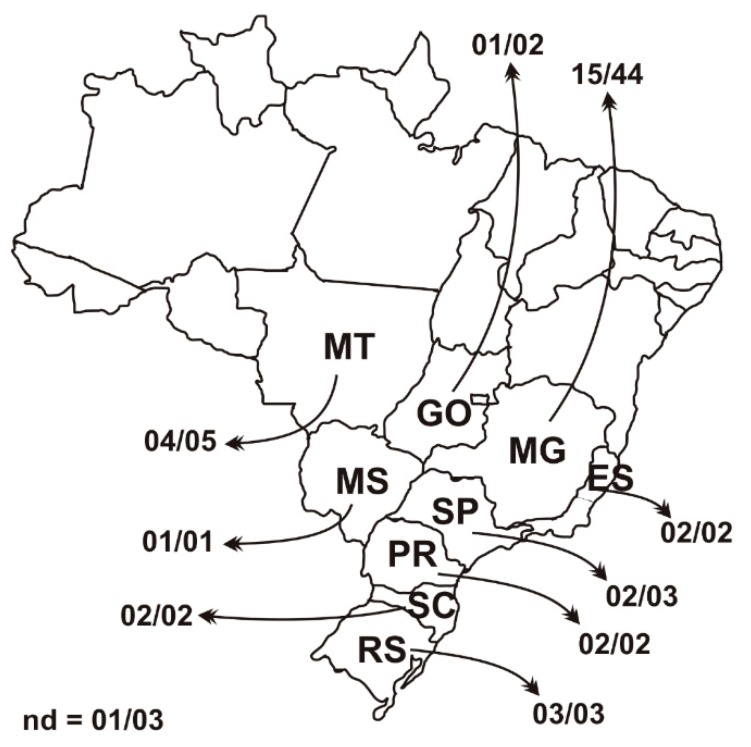
Distribution of the 67 tissue samples collected from pigs between 2006 and 2007 from nine states in Brazil. The number before the bar indicates the total of positive samples and the number after the bar indicates the sampling in each state. Nd = no data about sample origin; MT = Mato Grosso; MS = Mato Grosso do Sul; GO = Goiás; MG = Minas Gerais, SP = São Paulo; ES = Espírito Santo; SC = Santa Catarina; RS = Rio Grande do Sul; PR = Paraná.

**Figure 2 viruses-11-00201-f002:**
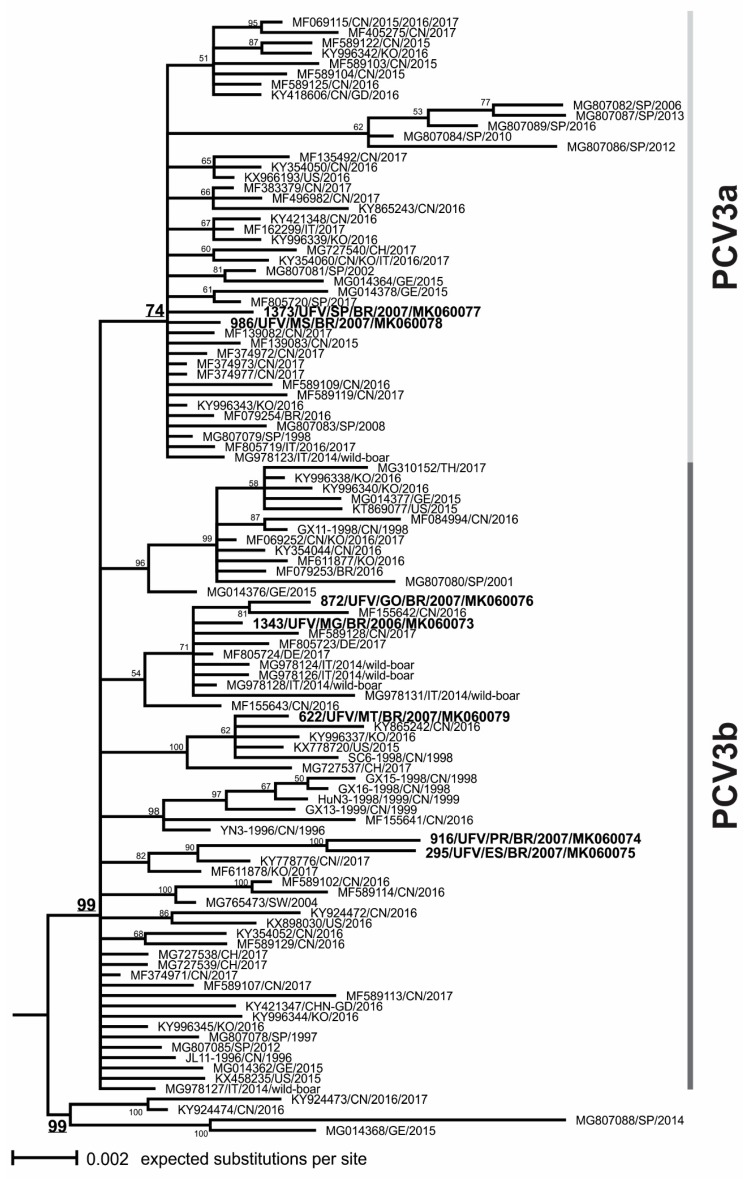
Phylogenetic clustering of PCV3 strains. The midpoint rooted majority-rule consensus tree was obtained by Bayesian Inference (BI) analysis of 108 ORF2 nucleotide sequences. The scale bar indicates nucleotide substitutions per site. The posterior probability values (expressed as percentages) are shown beside each node. The sequences obtained in this study are highlighted in bold.

**Table 1 viruses-11-00201-t001:** Polymorphisms identified in the Porcine circovirus 3 (PCV3) Cap protein sequences of Brazilian strains from 2006 and 2007 and those recently collected in 2016. Each column corresponds to the positions of amino acid residues which are variable.

GenBank Accession/Year	24	27	75	77	100	104	150
MF079253/2016	A	R	A	S	T	F	L
MF079254/2016	V	K	A	S	T	F	I
MK060073/2006 †	A	R	A	G	T	Y	I
MK060074/2007	A	R	S	S	S	F	L
MK060075/2007 †	-	-	A	S	T	F	L
MK060076/2007	-	-	A	S	T	Y	I
MK060077/2007	-	-	A	S	T	F	I
MK060078/2007 †	-	-	A	S	T	F	I
MK060079/2007	-	-	A	S	T	F	I

Subtitle “-” no substitution, when compared PCV3 sequences from GenBank. “†”: Co-infection with PCV2.

**Table 2 viruses-11-00201-t002:** Neutrality tests of ORF2 sequences.

	Tajima’s D	*p* *	Fu and Li’s D	*p* *	Fu and Li’s F	*p* *
**ORF2**	−2.25128	< 0.01	−4.76831	< 0.02	−4.48081	< 0.02

* Significant values: *p* < 0.05.

**Table 3 viruses-11-00201-t003:** Codon selection pressures on the sequences of ORF2.

Method	Number of Codons	Global dN/dS	Neutral Sites *	Negatives Selected Sites *	Codons
**SLAC**	214	0.1549	193 (90.19%)	21 (9.81%)	14; 21; 41; 49; 54; 55; 57; 70; 73; 82; 85; 95; 96; 129; 134; 147; 165; 192; 203; 212; 213
**FEL**	214	-	177 (82.71%)	37 (17.29%)	9; 11; 14; 16; 21; 26; 32; 38; 41; 49; 53; 54; 55; 57; 70; 73; 82; 85; 95; 99; 106; 118; 129; 134; 147; 162; 165; 166; 171; 176; 190; 192; 199; 203; 211; 212; 213
**iFEL**	214	-	204 (95.33%)	10 (4.67%)	9; 14; 49; 57; 70; 75; 85; 129; 134; 147

* Significant values: *p* < 0.05.

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
