# Peer review of "Retrospective Detection and Genetic Characterization of Porcine circovirus 3 (PCV3) Strains Identified between 2006 and 2007 in Brazil"

_viruses, 2019, doi:10.3390/v11030201_

Round 1
Reviewer 1 Report
This manuscript is a useful contribution to the field of PCV research, adding to the worldwide evidence base regarding PCV3 epidemiology.
Specific comments:
Line 81 - It would be useful to list the types of tissues used in this study.
Line 84 - A summary of the DNA extraction method should be included here.
Lines 92-93 - This isn't really part of the materials and methods, perhaps move to the discussion section.
Lines 138-139 - There is no mention in the materials & methods section of testing for PCV2. Please add this.
Lines 145-146 - Please explain why only 7 sequences were generated from 32 positive samples, and characterise which samples were successful for sequencing (i.e. from which states, with which clinical situation, etc.).
Lines 185-186 - How was the decision made regarding what constitutes a sub-type? Was a specific cut-off value for similarity used? This can also be used in the discussion section, lines 262-264.
Author Response
Dear Reviewer,
Thanks for your attention. The answers for comments are in the file attached.
Best wishes

Reviewer 2 Report
The manuscript on the retrospective analysis of PCV3 infection in farmed swine in Brazil is interesting and adds to the knowledge on an important topic - the pathogenicity of PCV3. The data supports the authors conclusions that PCV3 is not always associated with disease. However, the authors should add to their discussion/conclusion that the data is from 2006 - 2007 and the PCV3 virus could have evolved in the last one decade. They could relate the PCV3 evolution to the evolution of PCV2 as a pathogenic virus in the discussion as the two viruses are similar in their genetic organization. In this regard, analysis of current PCV3 prevalence in Brazil and ORF2 phylogenetics to those in 2006/07 would make the study more interesting. Further, as the authors have mentioned, analysis of PCV3 in wild boars would also add to the understanding.
In addition the authors could consider a few corrections to their manuscript as listed below.
Line 28: "Interestingly, no clinical signs were noticed in samples which were detected singular PCV3 infection" could be rephrased to "....no clinical signs were associated with samples which were......."
Lines 66 and 67: ".....over the past 50 years in the pork industry." should be rephrased to "...over the past 50 years in farmed pigs.".
Line 100: "....were submitted to electrophoresis..." should be rephrased as "....were subjected to electrophoresis..."
Line 214 - 215 is not clear. It should clearly specify that 37.5% (12/32) of the PCV3 positive samples are also positive for PCV2 and only 29.8%(20/67) of all samples were singly PCV3 positive. The percentage values are based of different sets of the population,i.e, PCV3 positive vs all samples.
Line 238/239: "....clinical manifestation of disease as PCV2 pathogenesis." should read as "....clinical manifestation of disease similar to PCV2 pathogenesis." (if this is what the authors want to communicate).
Author Response

(The authors gave the same response as above.)

Reviewer 3 Report
The manuscript of Saraiva et al., provides a certain insight into the PCV-3 epidemiology of PCV-3 in Brazil.
The study seems well planned and performed and the applied methodology adequate.
The main flaw of the study is that it is not structured to provide new insight on PCV3 biology of clinical relevance. However, these limits are recognized in the discussion and the authors are actually caution in their conclusion, avoiding any overstatement , which is commendable.
Globally, the results could be of interest because of the current limited availability of molecular epidemiology on this recently emerged virus.
I have just some doubts that should be clarified and some minor suggestion.
In M&M line 112: 7 Brazilian sequences were obtained. The reason why a just a subset of samples was obtained should be reported (low sequencing efficiency or sample selection ). Additionally brief (The row data are already reported as supplementary table, but a brief description could be of help) description of these samples should be provided since the main focus of the manuscript deals with molecular epidemiology data and the sampling features/geographic distribution could affect the representativeness of the provided information.
Line 119: 1% burn-in is reported. Although theoretically possible (if supported by MCMC trace examination) it sounds a little odd. Generally, at least 10% of the run is removed as burn-in. This could be a typo.
Results:
Was any statistical test performed to evaluate the association between PCV-3 and clinical signs and its interaction with PCV-2?
Table 1. the first column should probably be 24 instead of 244
several cells have a “-”. What is the meaning? Is due to the partial sequencing? If so this should be declared. However, if possible, efforts should be done to get the whole sequence of the considered region.
Discussion.
Line 230. “no clinical signs of infection”… I believe that “no clinical sign in PCV-3 positive and PCV-2 negative swine were noticed” could be more correct.
Line 231 “Although these results were found”. This sentence sounds strange in English. Overall, I’d suggest a revision from a native speaking person since there are some typos and some sentences appear a direct translation from a different language. However, being not a native speaker, I can not criticize/correct the language with the proper confidence.
Line 238: “as PCV-2 pathogenesis” should be “as already proven for PCV-2 pathogenesis.”
Line 272: The sentence should be: “These results confirms that both PCV3a and PV3b genotype...”
line 281: In the Italian manuscript the authors report :”Both clades included sequences sampled from commercial pigs in Italy, supporting some strain interchange between wild and domestic
swine populations”. Therefore the statements herein provided is not correct.
Line 289: This sentence should be “no clinical signs were noticed in samples where PCV-3 only was detected.”
Line 298:” PCV3 strains clustered in both PCV3a and PCV3b...”
p { margin-bottom: 6.25px; line-height: 120%; }
Author Response

(The authors gave the same response as above.)

Round 2
Reviewer 1 Report
Reviewer suggestions from the previous evaluation have all been addressed in the revised manuscript, resulting in an improved paper suitable for publishing.
Author Response
Dear Reviewer 1,
Thanks for your suggestions during the revisions, it helped us to improve our paper.
Best regards,
Abelardo Silva Júnior

Reviewer 2 Report
The authors have well documented the prevalence of PCV3 in Brazil from 2006-2007 and associated trends such as genetic variation and disease association. The manuscript is certainly of interest to a wider audience given to the importance of PCV3 in the current scenario. The authors should followup this article with more information linking the current PCV3 circulating strains in the field with those from 2006-2007. This will inform us about the evolution of PCV3 in Brazil over the last decade.
Author Response
Dear Reviewer 2,
We would like to thank you all for carefully reading this manuscript and for the thoughtful comments and constructive suggestions. We consider all your comments during the revisions. And we did the corrections as you can see in the manuscript, which helped us to improve its quality.
1- The authors have well documented the prevalence of PCV3 in Brazil from 2006-2007 and associated trends such as genetic variation and disease association. The manuscript is certainly of interest to a wider audience given to the importance of PCV3 in the current scenario. The authors should follow up this article with more information linking the current PCV3 circulating strains in the field with those from 2006-2007. This will inform us about the evolution of PCV3 in Brazil over the last decade.
Thanks for your comment. It is very important for us this explaining. We made a link between PCV3 sequences from 2006-2007 and the sequences deposited recently in GenBank. Our results indicate that PCV3 Brazilian strains probably showed genetic stability over the past 10 years. You can see the explanation in lines 179-187 (table 2 and 3).
“The tests of neutrality suggest that the evolution of PCV3 ORF2 gene possibly occurred under the influence of selection and that a purifying selection took place (Table 2), considering the significance and the high negative values of calculated coefficients. The dN/dS ratio is also a versatile way to predict the action of natural selection on gene sequences, once it provides an estimate of selection pressure at the protein level (Kimura, 1983). A global dN/dS ratio of 0.155 was calculated for the ORF2 sequences and up to 17% of its codons were predicted to be evolving under a purifying selection, which indicates a strong purifying selection is acting over the Cap protein (Table 3). Therefore, Cap protein of PCV3 seems to be subjected to strong restrictions and unable to tolerate high levels of variation on its sequence.”
In the discussion section you can also see in lines 267-271:
“Our PCV3 sequences collected in 2006 and 2007 shared high identity (97.94% to 99.62%) with the Brazilian strains obtained in 2016 [13], indicating the genetic stability of PCV3 Brazilian strains over the past 10 years. This finding can be corroborated by the neutrality and selective pressure test results, in which the Cap protein of PCV3 seems to be unable to tolerate high levels of variation on its sequence.”
Also, in conclusion section in lines 302-306:
“The estimation of selective pressures indicated that PCV3 Cap protein seems to be evolving under a strong purifying selection, being unable to tolerate high levels of variation on its amino acid sequence. Furthermore, PCV3 sequences collected in 2006 and 2007 shared high identity with the Brazilian strains obtained in 2016. These results indicate that PCV3 Brazilian strains probably showed genetic stability over the past 10 years.”
Best regards,
Abelardo Silva Júnior

Reviewer 3 Report
I believe that the manuscript is technically sound for publication, I have just some minor points.
Caption of Table 1: "Subtitle “-“ no substitution." no substitution compared to who? the first strain? the reference should be reported
"line 277: "These results confirms that both PCV3a and PV3b genotype booth PCV3a and PCV3b genotypes". Part of the previous sentence has not been deleted.
Was any statistical test performed to evaluate the association between PCV-3 and
clinical signs and its interaction with PCV-2?
Answer: The analysis binary data of Table S1 didn’t found any correlation between PCV-3 and
clinical signs and its interaction with PCV-2."
I appreciate that the authors performed the analysis, however it think it would be important to explicitly report the absence of any statistical association in the manuscript!
Author Response
Dear Reviewer 3,
We would like to thank you all for carefully reading this manuscript and for the thoughtful comments and constructive suggestions.
We consider all your comments during the revisions. And we did the corrections as you can see in the manuscript, which helped us to improve its quality.
1- I believe that the manuscript is technically sound for publication, I have just some minor points:
1.1 Caption of Table 1: "Subtitle “-“ no substitution." no substitution compared to who? the first strain? the reference should be reported
Thanks for your comment. The Brazilian PCV3 sequences were compared with the set of sequences of PCV3 downloaded from GenBank. We added this comment in line 166.
1.2 "line 277: "These results confirms that both PCV3a and PV3b genotype booth PCV3a and PCV3b genotypes". Part of the previous sentence has not been deleted.
You are right. We committed this mistake but now it was corrected. Thanks for your comment.
1.3 Was any statistical test performed to evaluate the association between PCV-3 and
clinical signs and its interaction with PCV-2?
Answer: “The analysis binary data of Table S1 didn’t found any correlation between PCV-3 and
clinical signs and its interaction with PCV-2."
I appreciate that the authors performed the analysis, however it think it would be important to explicitly report the absence of any statistical association in the manuscript!
We agreed with your comment and we added a phrase about that in lines 144-145.
Best regards,
Abelardo Silva Júnior
